# Genome-Wide Association and Prediction of Traits Related to Salt Tolerance in Autotetraploid Alfalfa (*Medicago sativa* L.)

**DOI:** 10.3390/ijms21093361

**Published:** 2020-05-09

**Authors:** Cesar Augusto Medina, Charles Hawkins, Xiang-Ping Liu, Michael Peel, Long-Xi Yu

**Affiliations:** 1United States Department of Agriculture-Agricultural Research Service, Plant Germplasm Introduction and Testing Research, Prosser, WA 99350, USA; cesar.medinaculma@wsu.edu (C.A.M.); hawkcharles@gmail.com (C.H.); liuxiangping3885@gmail.com (X.-P.L.); 2Current address: Department of Plant Biology, Carnegie Institution for Science, Stanford, CA 94305, USA; 3Current address: College of Animal Science & Veterinary Medicine, Heilongjiang Bayi Agricultural University, Daqing 163316, Heilongjiang, China; 4United States Department of Agriculture-Agricultural Research Service, Forage and Range Research Lab, Logan, UT 84322, USA; mike.peel@usda.gov

**Keywords:** association mapping, genomic selection, GBS, allele dosage, abiotic stress, polyploid

## Abstract

Soil salinity is a growing problem in world production agriculture. Continued improvement in crop salt tolerance will require the implementation of innovative breeding strategies such as marker-assisted selection (MAS) and genomic selection (GS). Genetic analyses for yield and vigor traits under salt stress in alfalfa breeding populations with three different phenotypic datasets was assessed. Genotype-by-sequencing (GBS) developed markers with allele dosage and phenotypic data were analyzed by genome-wide association studies (GWAS) and GS using different models. GWAS identified 27 single nucleotide polymorphism (SNP) markers associated with salt tolerance. Mapping SNPs markers against the *Medicago truncatula* reference genome revealed several putative candidate genes based on their roles in response to salt stress. Additionally, eight GS models were used to estimate breeding values of the training population under salt stress. Highest prediction accuracies and root mean square errors were used to determine the best prediction model. The machine learning methods (support vector machine and random forest) performance best with the prediction accuracy of 0.793 for yield. The marker loci and candidate genes identified, along with optimized GS prediction models, were shown to be useful in improvement of alfalfa with enhanced salt tolerance. DNA markers and the outcome of the GS will be made available to the alfalfa breeding community in efforts to accelerate genetic gains, in the development of biotic stress tolerant and more productive modern-day alfalfa cultivars.

## 1. Introduction

The impacts of soil salinization on world agriculture will become more pervasive and severe in the future. Quadir et al. [1] estimated that global soil salinity costs $27 B in lost agricultural productivity per year, and the extent of saline soils are increasing. Salinization of soils can occur as a result of natural processes (primary salinization) or as a result of human activities (secondary salinization). In areas where the water table is near the surface, a continuous column of water can form between the surface and the (saline) water table. When this occurs, evapotranspiration at the surface creates a “wicking” effect that continuously draws more water to the surface. As surface water is lost, salts precipitate and remain in the topsoil layers. Irrigated agriculture can also cause salt levels to increase over time, mainly from use of high-salt irrigation water. This problem is exacerbated in areas with poor drainage. High soil salt levels can be managed by leaching, which involves the application of excess irrigation water to dissolve salt and carry it away via leaching [2]. The required leaching fraction (proportion of excess water) decreases with more salt-tolerant plants. Therefore, increasing a crops’ salt tolerance can potentially reduce water usage, irrigation costs, and environmental impacts.

Salt affects plant growth indirectly by its negative effect on soil water potential and by direct toxicity when it is taken up by the plant. In the former, the increased ion concentration in the soil decreases the soil water potential, which makes it more difficult for the plant to access water. Therefore, salt stress affects plants in a similar manner to drought stress with shared physiological mechanism. These mechanisms include stomatal closure, decreased rates of photosynthesis, formation of reactive oxygen species (ROS), decreased water content in plant cell and attendant problems with protein folding [3]. Consequently, mechanisms of plant resistance to this form of salt stress are similar to those in drought tolerance. Both salt and drought stress resistances involve production of compatible osmolytes to decrease cell water potential and draw in more water, increased production of heat-shock proteins (HSPs) and other chaperones to improve the correct protein folding, and production of antioxidants to quench ROS [4,5].

The second salt stress mechanism, direct toxicity, is thought to be a result of sodium (Na^+^) accumulation within the cell and the homeostasis between Na^+^ and potassium (K^+^) [2]. Sodium toxicity interferes with the uptake of other cations such as calcium (Ca^++)^ and potassium (K^+^) ultimately resulting in reduced growth, leaf chlorosis, and early leaf senescence. Cross talk between gene regulatory networks attributed to drought and salt stresses has been found. It has been suggested that a combined effect of dehydration and osmotic stress may cause greater regulation in plant response to salt stress (see review [3]). Functional genomics provides a new tool to address the genetic bases and physiological mechanisms of plant salinity tolerance.

Cultivated alfalfa is an outcrossing autotetraploid (2n = 4x = 32) species with a genome size of 800−1000 Mb [6]. Alfalfa genetic improvement to salt tolerance has been limited in part due to the genetic complexity of the trait which is under polygenic control and interacted with environmental factors [4]. Breeding of alfalfa is complicated by its tetraploid genome and by its out-crossing pollination which prevents the creation of inbred lines.

Alfalfa cultivar development efforts have largely focused on the phenotypic selection in field environments. Recurrent selection is used for improving traits of interest in a quantitative manner. The strategy is to gradually increase the frequency of favorable alleles and maintain genetic variability for future selection. Progress on the improvement of the traits under recurrent selection are made, but it takes long periods to successfully develop new varieties. Recurrent selection methods would be most effective when integrated with marker-assisted selection (MAS) [7]. MAS is a procedure for selecting traits of interest based on DNA markers linked to quantitative trait loci (QTL). QTLs are detected through genetic mapping or genome-wide association studies (GWAS) where the QTL signals above specific thresholds are declared statistically significant. However, in complex traits (e.g., stress tolerance or yield) it is often not possible to clearly identify QTL or multiple loci distributed throughout the genome and work in concert to control the trait.

Genomic selection (GS) is a promising alternative to phenotype-based selection of crops in breeding programs. The objective of GS is to determine the genetic potential of an individual based on whole genome markers, instead of focusing of specific QTL. Therefore, GS does not depend on prior knowledge about QTL effects. This technique is based on the association of phenotypic traits with genome-wide markers to obtain the genomic estimated breeding values (GEBV) [8]. GEVB are obtained by training statistical or machine learning methods. Predictive trained models are then applied to identify the best individuals in testing populations, based solely on their genotypic profiles. GS is used to intensify the selection process by increased selection efficiency or by reducing selection cycles. In this way GS reduces the cost per cycle and the time required for variety development [9].

In GS, several statistic models have been used for predicting breeding values. Support vector machines (SVM) and random forest (RF) are supervised machine learning methods used to predict the target phenotype $\left(y_{i}\right)$ in which training datasets with large number of predictors (Markers or $g\left(x_{i}\right)$) $(y_{i}\sim \mu +g\left(x_{i}\right)+e$) are used. These methods are based on the identification of an objective function and its optimization [10]. The objective function has two parts: (1) training loss function and (2) regularization term. The first part tests how well a model fits on a training dataset (presented as root mean square error (RMSE)), while the second part measures the complexity of the model as more complex models produce more unstable results [11]. These supervised machine learning methods can handle high dimensionality problems (p≪n) where the p:n ratio exceeds 50–100 [12] and they do not assume a priori linear and additive action of markers.

The objectives of this work were to use GWAS and GS methods to identify loci associated with salt tolerance and to predict breeding values using single nucleotide poplymorphism (SNP) markers with allele dosage in breeding populations of autoteraploid alfalfa. Agronomic traits such as biomass yield and plant growth vigor under salt stress were evaluated in the field. Genome-wide DNA markers were developed using genotype-by-sequencing (GBS) and used for GWAS and GS. Six statistic models were used in GWASpoly to identify loci associated salt tolerance and eight genomic prediction models were tested on the prediction accuracy for GEBV in the breeding populations toward improving salt tolerance in alfalfa.

## 2. Results

### 2.1. Coverage and Marker Density

Of the 240,444,007 raw reads obtained from the population via GBS, Bowtie2 successfully aligned 91,360,439 reads one time (38.0%) and 100,635,037 reads multiple times (41.8%) to the *M. truncatula* genome v5.0. After filtering, 6862 high quality biallelic single nucleotide variants (SNVs) were obtained and annotated using the functional annotation of variants module of Next Generation Sequencing Experience Platform (NGSEP). The biallelic SNVs were annotated as follows: 5234 markers as protein-coding loci (76.8%) and 1628 markers as non-coding loci (23.7%) (Table 1). The distributions of allele frequency were 40.0% between 0.05 and 0.1; 23.2% between 0.1 and 0.2; 14.76% between 0.2 and 0.3; 11.8% between 0.3 and 0.4; and 10.2% between 0.4 and 0.5 (Figure 1A). The distributions of markers by chromosomes were as follows: Chr. 1 = 1056 markers, Chr. 2 = 900 markers, Chr. 3 = 1145 markers, Chr. 5 = 822 markers, Chr. 6 = 505 markers, Chr. 7 = 783 markers, Chr. 8 = 788 markers, and 36 markers located into contigs without chromosome assignment. The high-quality GBS markers were plotted according to their position in the chromosomes of *M. truncatula* v5.0. The distribution of the markers across the chromosomes was not uniform and presented gaps in coverage towards the inner part of some chromosomes due to possible centromeric regions (Figure 1B). Finally, biallelic SNVs were transformed into GWASpoly format with NGSEP software v 3.3.3 and were subjected to GWAS and GS analysis. The GWASpoly allowed identifying the allele dosage in tetraploid genotypes with up to five alleles at each locus [5]. The allele frequency was plotted against the allele type in Figure 2. The frequencies of five major alleles were AAAA = 0.42, AAAB = 0.15, AABB = 0.19, ABBB = 0.08, and BBBB = 0.14 (Figure 2).

### 2.2. Genome-Wide Association Studies

GWAS were performed using the combination of phenotypic data on vigor and yield from the 2018 and 2019 field evaluations and genotypic data with allele dosage. GWAS analysis of vigor identified 21 markers at 16 loci in evaluations from the two fields sites using general and diplo-general models. Six of these markers (chr. 1 50528093 and 50528125, chr. 2 35034036, chr. 4 44369334, chr. 5 41782228, and chr. 7 26012100) were identified in populations evaluated at the Castle Dale, Utah site (Figure 3A,B). A total of 15 markers were identified (chr. 1 19123928, chr. 2 44365722, chr. 3 2641319, 2641320, 49957218, and 49957253, chr. 5 12453276, 12453319, 12453328, and 35355162, chr. 6 7243498, 35426314, and 40502777, and chr. 7 43123906 and 44707092) for the Othello, Washington site (Figure 3C,D). The loci identified by GWASpoly were aligned to the corresponding genomic region using the *M. truncatula* genome v5.0 as reference. Of 16 loci identified, 14 were targeted to the coding regions of protein loci (Table 2). The protein-coding loci were annotated as follows: MtrunA17_Chr1g0205221 was annotated to folate-biopterin transporter, major facilitator superfamily domain-containing protein; MtrunA17_Chr2g0324021 to oxidoreductase; MtrunA17_Chr3R0014140 to RLX_singleton_family134; MtrunA17_Chr4g0048811 to aminoacyltransferase, E1 ubiquitin-activating enzyme; MtrunA17_Chr5g0410771 to HSP20-like chaperone, P-loop containing nucleoside triphosphate hydrolase; MtrunA17_Chr5g0435221 to putative 23S rRNA (adenine(2503)-C(2))-methyltransferase; MtrunA17_Chr5g0444321 to leucine-rich repeat domain, L domain-containing protein; MtrunA17_Chr6R0226110 to RLG_singleton_family376; MtrunA17_Chr6g0486011 to zinc finger, RanBP2-type; MtrunA17_Chr7g0235641 to putative RIN4, pathogenic type III effector avirulence factor Avr cleavage; MtrunA17_Chr7g0259771 to small GTPase superfamily, EF-hand domain pair.

The GWAS identified six markers significantly associated with yield under salt stress. These markers were chr2_8865320, chr3_5484686 and 17906891, chr4_54035230, chr6_1909362, chr 8_32682521. Two of these markers were associated with the yield in 2018 field evaluations. Among them, marker chr3_5484686 was identified by 2-dominant reference model and chr6_1909362 was identified by general model (Figure 4A,B). One marker (chr3_17906891) was identified for yield from the July 2018 harvest by the diplo-general model (Figure 4C). Marker chr6_1909362 was identified by the general model in both August and September 2018 harvests (Figure 4D,E). Two yield markers were identified from the June 2019 harvest. Among them, marker chr. 2 38865320 was identified by the diplo-general, diplo-additive, and 1-dominant reference models while marker chr. 8 32682521 was identify by diplo-general model (Figure 4F–H). Yield marker chr. 2 38865320 was identified in the July 2019 harvest by the diplo-general, diplo-additive, and 1-dominant reference model (Figure 4I–K). Marker chr. 4 54035230 was identified from yield data for the September 2019 harvest by 2-domimant reference model (Figure 4L). It is noteworthy that marker chr. 2 38865320 was associated in both the June and July 2019 harvests and marker chr. 6 1909362 was associated to harvests in August and September along with the total yield of 2018. However, May 2019 yield and total yield during 2019 did not show any associated markers with the six models tested.

The six markers identified were annotated to their genomic regions using the *M. truncatula* genome v5.0 as reference and all markers were targeted to protein-coding loci (Table 2). The protein-coding locus MtrunA17_Chr2g0316741 was annotated to hypothetical protein; Chr3g0083861 to serpin family protein; MtrunA17_Chr3g0094791 to tetratricopeptide-like helical domain, DYW domain-containing protein; MtrunA17_Chr4g0062111 to chaperone-like protein of POR1; MtrunA17_Chr6g0451341 to transcription regulator IWS1 family; MtrunA17_Chr8g0369441 to brevis radix (BRX) domain, transcription factor BREVIS RADIX domain-containing protein (Table 2).

### 2.3. Linkage Disequilibrium Analysis

Linkage disequilibrium (LD) analysis was performed with all markers associated with yield and vigor under salt stress and their adjacent markers in a 10 kb window by Haploview v4.2 [14]. Among 27 markers identified for yield and vigor, six blocks were identified to harbor multiple markers at the same locus including block 1 on chromosome 1 at the positions 50527909, 50528082, 50528093 and 50528125; block 2 on chromosome 2 at positions 44365722, 44365739, 44365748, and 44365762; block 3 chromosome 3 at positions 5484625, 5484632, 5484637, and 5484686; block 4 on chromosome 4 at positions 44369328, 44369331, and 44369334; block 5 on chromosome 5 on positions 12453276, 12453319, and 12453328; and Block 6 on chromosome 8 at positions 32682474 and 32682521 (Figure 5).

### 2.4. Genomic Selection

The growth vigor under salt stress collected in Othello and Castle Dale and yield collected in Othello were used for GS using eight different models: rrBLUP, BayesA, BayesB, BayesC, BRR, BL, SVM, and RF. GS used 10-fold cross validation between a training population of 90% and a testing population of 10% to predict breeding values. The accuracy of Pearson’s correlation between predicted GEVB and phenotypic values was used in all datasets. Mean accuracies for the eight models tested were 0.264 (SD ± 0.015) in Castle Dale and 0.337 (SD ± 0.011) in Othello, and mean RMSE values were 0.889 (SD ± 0.005) in Castle Dale and 0.6962 (SD ± 0.005) in Othello. The best fitting model was SVM in both datasets with accuracies of 0.287 in Castle Dale and 0.361 in Othello (Table 3).

The prediction accuracy, based on Pearson’s correlations, varied across harvest dates for the yield trait. The highest prediction accuracy was obtained for the September 2018 harvest data with mean accuracy of 0.457 (SD ± 0.021) for all models (data not shown). The lowest prediction accuracy was found in harvest data for All 2019 with a mean accuracy of 0.087 (SD ± 0.031) for all models. Harvests in August 2018, All 2018, July 2019 and September 2019 had similar mean accuracies of 0.262 (SD ± 0.011), 0.239 (SD ± 0.030), 0.254 (SD ± 0.021), respectively (Table 4). The range of means by models in all yield datasets were from 0.224 (SD ± 0.112) for BayesA to 0.275 (SD ± 0.095) for RF.

To analyze the variation in the errors in a set of forecasts, the mean absolute error (MAE) and the root mean squared error (RMSE) were used to measure the average magnitude in the continuous variable errors (i.e., yield). Comparisons between the models and datasets by MAE and RMSE identified high correlations (Appendix A) therefore only RMSE was used to test the models. Comparisons of accuracy (Pearson’s correlation) and RMSE indicated that SVM was the best fitting model for yield data in September for both 2018 and 2019, while the RF model fit the data best for yield in July 2018, May 2019, June 2019, and July 2019 (Table 4). Different parameter tunings were tested to achieve the lowest RMSE value (Appendix A). The costs (C) of parameter tuning SVM {0.25,0.5,1.0} were used to control the trade-off between smooth decision boundary (hyperplane) that classifies the training predictors correctly and sigma (σ) that defines how far training predictors influence regression. High σ values only consider the closest predictors to the hyperplane while low values consider the influences of all predictors. SVM had common cost of 1 in almost all datasets and σ values were between 0.000098 and 0.00012. Parameters adjusted in RF were randomly chosen subset of M (predictor variables SNPs) for determining a decision tree and split rule which defines the kernel (“variance” or “extra-trees”) to split the candidate variables (predictor variables) that minimizes the sum of squared estimate of errors (SSE). Parameter “variance” was used for splitting rule for the yields of July 2018, September 2018, May 2019, and September 2019 while “extra-trees” were used for August 2018, June 2019, and July 2019. The most frequent mtry value was 6832, which correspond to the complete set of SNPs (Appendix A).

Finally, in order to test machine learning models, 10% of the dataset was left-out during the model’s training by 10-fold cross-validation. Finally, the model trained was used to predict yield of the 10% dataset left-out comparing the goodness of fit of the models by accuracy and RMSE values of model trained and model with future data (Table 5). This approach allowed to increase the accuracy in eight of nine datasets. By this approach the accuracy in the July 2018, August 2018, May 2019, June 2019, July 2019, September 2019, and All 2019 datasets was increased for the SVM model. Maximum values were found in September 2018 with and accuracy of 0.771 for the RF model and in July 2018 with an accuracy of 0.793 for the SVM model (Table 5).

## 3. Discussion

### 3.1. GBS and Allele Dosage

Estimation of allele dosage is crucial for precise GWAS and GS analyses in polyploid species. Pipelines using diploidization of polyploid makers could affect GWAS [16] or GS [17] results. Therefore, using a pipeline that accurately includes allele dosage can improve genotyping accuracy and reduce errors in polyploid species. In this study the software NGSEP v4.0.0 was used to analyze allele dosage and to obtain 6862 meaningful variants. The number of markers in this study was similar to previous reports using the diploidization pipelines in alfalfa [18]. Markers with allele dosage in the present study allowed performing GWAS with the GWASpoly software which was originally designed for association mapping in the autotetraploid species potato [5].

### 3.2. Association Mapping

Using GWASpoly, we identified 27 SNPs associated with salt stress tolerance. Among them, six were associated with yield and 21 were associated with vigor under salt stress. Of the 27 markers identified, three were found to be in non-coding regions and four were associated with hypothetical proteins. The 20 remaining markers were associated with 16 protein-coding loci annotated with known functions. Locus MtrunA17_Chr1g0205221 was associated to a putative folate-biopterin transporter. The folate-biopterin transporter (FBT) belongs to the major facilitator superfamily (MFS). Some members of this family are Zinc-Induced Facilitator-Like 1 (ZIFL1) proteins that have reported activity as polar auxin transport modulators and alternative splicing for drought tolerance [19]. FBT has been involved in transport of organic molecules (e.g., folate) containing nitrogen [20]. In the present study, four SNPs were associated with the gene MtrunA17_Chr1g0205221 which showed 84% similarity to the FBT protein At2g32040.2 in *Arabidopsis thaliana* (https://medicago.toulouse.inra.fr/MtrunA17r5.0-ANR/). Mutation of At2g32040.2 in *A. thaliana* increased the total chloroplast folate content and decreased the proportion of 5-methyl-tetrahydrofolate [21]. Increased folate levels have been associated with germination and vigor in barley under salt stress [22]. Kılıç and Aca (2016) found that exogenous application of folic acid was involved in mitigation of salt-induced inhibition, and reduced the negative effects of salt on barley germination. These observations agree with current findings where the SNPs 50527909, 50528082, 50528093, and 50528125 located at the MtrunA17_Chr1g0205221 locus were associated with plant vigor under salt stress.

Locus MtrunA17_Chr2g0324021 was annotated as putative oxidoreductase in the short-chain dehydrogenase reductase (SDR) class. SDR proteins are involved in oxidative reduction affecting multiple metabolic processes. According to *M. truncatula* genome browser [13] MtrunA17_Chr2g0324021 has 61.1% identity with SDR5 in *A. thaliana* and 78.3% and 80.3% similarity to 3-beta-hydroxy-Delta(5)-steroid-dehydrogenase in green bean (*Phaseolus vulgaris*) and soybean (*Glycine max*) proteomes, respectively. SDR5 belongs to a NAD(P)-binding Rossmann-fold superfamily protein which have been shown to be induced with Methyl jasmonate and reduce the effects of abiotic stresses in plants [23].

Locus MtrunA17_Chr3R0014140, associated with 2641319 and 2641320 SNPs, was annotated to an RLX_singleton_family134 and a domain search in interproscan [24] predicted a PWWP domain-containing protein which is a structural module characteristic of chromatin regulators. Proteins with PWWP domain are involved in histone interactions affecting development and flowering time of *A. thaliana* [25]. Additionally, Waidmann et al. [26] reported DEK3, a protein with a PWWP domain downregulated by salt stress in roots and shoots in *A. thaliana*. The process of acetylation and methylation of histones in response to salt stress control the ABA signaling process which play an essential role in organ to organ communication [27]. Similarly, locus MtrunA17_Chr3g0094791 associated with SNP 17906891 that was associated with a putative tetratricopeptide-like helical domain, DYW domain-containing protein. This protein has been shown to be involved in abscisic acid responses and osmotic stress tolerance [28]. Furthermore, domain DYW has a role in RNA editing in plant mitochondria in *A. thaliana* [29] rice (*Oryza sativa*) [30] and soybean [31]. Interestingly, in soybean the *GmPPR4* a DYW subgroup of pentatricopeptide-repeat (PPR) proteins was induced under salt and drought stresses [31].

Locus MtrunA17_Chr3g0083861 annotated to putative serpin family protein. Serpins act as protease inhibitors of serine proteases with other described roles in plant pathogen interactions [32], grain development in wheat (*Triticum aestivum*) [33], transport of RNA through phloem in response to biotic and abiotic stress [34] and drought stress tolerance [35]. The involvement of serpin in salt stress was also shown in proteomic analyses of wheat, where it was found that the overexpression of the protein Serpin Z1A in plants subjected to salt stress promoted plant growth through rhizobacterium *Enterobacter cloacae* SBP-8 [36]. The role of serpins in salt stress have been shown to limit protein degradation and reduced membrane degradation, ion leakage, senescence, and reactive oxygen species (ROS) induction by abiotic stresses [37].

Locus MtrunA17_Chr5g0410771 with three SNPs 12453276, 12453319, and 12453328 was annotated to a HSP20-like chaperone, P-loop containing nucleoside triphosphate hydrolase. HSP20-like chaperone is a stress responsive protein which is considered an early indicator of oxidative stress and ER stress. Previous reports has found HSP20-like chaperone upregulated by high salinity in Arabidopsis [38,39], rice (OsHSP20) [38], potato (StHsp20) [40], and poplar [41]. Abiotic stresses can cause protein aggregation or misfolding, therefore protective function of HSP20-like chaperone is crucial in plant response to salt stress.

The SNP 1909362 associated with yield in three different harvest was identified in the protein-coding loci MtrunA17_Chr6g0451341 annotated as Putative transcription regulator IWS1 family. IWS1 is a transcriptional regulator involved in brassinosteroid induced gene expression after its recruitment by BES1 in *Arabidopsis thaliana* [42]. These results agree with the role of brassinisteroids in reduce the deleterious effects caused by multiple abiotic stresses including salt stress [43,44,45]. This finding is significant because IWS1 TF affects the histone methylation to repress bassinosteroid induced gene expression. Additionally, IWS1 can repress transcription of NITRITE TRANSPORTER 2.1 in response to high nitrogen supply controlling the nutrient acquisition in plants and it has been proposed that this TF might affect distinct signaling pathways [46].

Locus MtrunA17_Chr6R0226110 was annotated as Putative potassium channel, voltage-dependent ERG. In mammalian Erg family voltage-gated K+ channels are specialized in repolarization of plateau potentials such as cardiac action potentials [47].

However, in plants, K+ channels have a fundamental role in homeostatic balance of the K+ [48]. In barley, better retention of K+ is related with salt-tolerant varieties because it helps to maintain the optimal cytosolic K+/Na+ homeostasis [49]. Additionally, MtrunA17_Chr6R0226110 has 51.2% with the protein AT3G17700.1 annotated as cyclic nucleotide-binding transporter 1 with a proved role in salt stress [50].

Locus MtrunA17_Chr6g0486011 was annotated as Putative RanBP2-type zinc finger protein. Zinc finger proteins are involved with the interaction with DNA, RNA, or proteins regulating in different plant processes like development and programmed death cell. The RanBP2-type zinc finger proteins are ssRNA-binding proteins with high affinity to RNA sequences containing a GGU motif [51]. Although, there are no reports of this class of zinc finger associated with salt stress, other classes of zinc finger such as CCCH-type or RR-type zinc finger proteins have been reported with significant roles in response to salt stress. The CCCH-type zinc finger proteins play important roles in regulation salt stress responses in Arabidopsis and mutations in the genes atszf1-1/atszf2-1 causes plants more susceptible to salt stress [52]. Finally the gene AtTZF3 classified as RR-TZF acts a negative regulator of seed germination under conditions of salt stress in wheat and Arabidopsis [53].

Locus MtrunA17_Chr8g0369441 was annotated as Putative brevis radix (BRX) domain, transcription factor. BRX domain-containing protein has been identified as a modulator of root growth in a dosage-dependent dominant negative effect [54] and it has been reported that this protein is involved in lateral root initiation which can be affected negatively by brassinosteroids and positively by auxins and cytokinins [55,56]. Root growth is a crucial factor for plant surviving under salt stress. Additionally, OsBRXL1, OsBRXL3 and OsBRXL4 homologous genes were expressed differentially under salt stress in rice [57].

Locus MtrunA17_Chr7g0235641 was annotated as Putative RIN4, pathogenic type III effector for the avirulence factor Avr cleavage. RIN4 has been described as one of the most important and best studied hubs involved in the regulation of two branches of plant immunity: PAMP triggered immunity and effector trigger immunity (reviewed in [58]). Additionally, it is known that RIN4 regulates stomata aperture by the interaction with plasma membrane H+-ATPases AHA1 and AHA2 in response to biotic stress [59] and with GENERAL CONTROL NONREPRESSIBLE4 (GCN4), an AAA+-ATPase family protein involving in regulation of stomatal aperture during abiotic stress by the degradation of RIN4 and 14-3-3 proteins to inhibit H+-ATPase activity [60]. Additionally, RIN4 also interacts with remorin protein, which increasese its transcription during salt stress [61]. This information shows the role of RIN4 in control of stomata aperture in biotic and abiotic stress.

Finally, locus MtrunA17_Chr4g0062111 annotated as Putative protein chaperone-like protein of POR1 (CPP1) (previously known as Cell growth defect factor 1), shows localization in mitochondria [62] and in plastids [63]. CPP1 has been found in a QTL associated with flowering date in Barley [64]. In plastids, CPP1 has an essential role in chloroplast development in *A. thaliana* and *Nicotiana benthamiana* regulating and stabilizing the function of light-dependent protochlorophyllide oxidoreductase (POR) [65]. CPP1 has a role controlling photo-oxidative stress caused by heat or ROS in chloroplasts and CPP1 deficiency produced etiolated seedlings. Additionally, it has been reported that downregulation of POR activity under salt stress affects the chloroplast biogenesis in rice [66].

The different markers associated in this study highlights the complexity of salt stress response and the multiple mechanisms of response to salt stress, which include control of protein degradation, chromatin modification, chaperon and TF gene activations, plant hormones signaling or homeostasis Na+/K+. However, there were some loci without a clear role in response to salt stress according to literature search. For example, locus MtrunA17_Chr7g0259771 annotated as Putative small GTPase superfamily, EF-hand domain pair, locus MtrunA17_Chr5g0435221 annotated as Putative 23S rRNA (adenine(2503)-C(2))-methyltransferase, or locus MtrunA17_Chr5g0444321 annotated as Putative leucine-rich repeat domain, L domain-containing protein.

### 3.3. Genomic Selection

Genomic selection has been significantly used in animal breeding over the past 15 years and has been applied to different crops as well (reviewed by Lin et al. [67]). Usually, Pearson’s correlation has been used to estimate prediction accuracy in GS in crops. However, Pearson’s correlation may not be the best choice when machine learning methods are used. In this work, we tested eight GS models according to accuracy, RMSE, and MAE and identified the correlation with different phenotypic parameters.

The RMSE approach is useful in GS when continuous variations of phenotypic values were used. It was used in genomic selection to avoid misselection of an appropriate prediction model [68]. In the present work we found a negative correlation between accuracy and RMSE (R = -0.64), which allowed to identify SVM and RF as the best models for predicting the breeding value using the high accuracies and low RMSE values. Other models such as rrBLUP or Bayesian methods (BayesA, BayesB, BayesC, Bayesian LASSO, and Bayesian ridge regression) did not show a significant performance among the traits tested. These results agree with previous reports where machine learning methods showed higher accuracies than those of other methods such as rBLUP or Bayes alphabet [69]. A previous report in alfalfa has also demonstrated that SVM was the best model in GS [18]. Additionally, the Caret package allows parameter tuning for machine learning methods based on a reduction of RMSE values and therefore finds the best parameters of the model (Appendix A).

The best performance of machine learning methods in GS in this work was likely due to the ability of these methods for identification of the top-ranking SNPs with major effects on the phenotypic variation and hence explained the large proportion of the additive genetic variance. Additionally, machine learning methods can capture complex SNP–SNP interactions and nonlinear relationships increasing the genetic variance and the heritability of the trait. Our results agree with the previous reports where supervised machine learning methods performed better when traits had dominant and epistatic effects [69].

The strategy of leave-out 10% of the individuals of cross-validation used in the present work allowed to test goodness of fit of the model in predicting the phenotypic traits in new testing individuals. In our analysis, the mean accuracy for all harvests increased from 0.275 to 0.377 with RF model and from 0.264 to 0.411 with SVM. Additionally, the RMSE values decreased from 0.426 to 0.412 with RF and there were no changes in SVM (0.425) (Table 5). The prediction accuracy increased in eight of nine harvests tested, reaching accuracies to 0.793 with SVM in the yield of July 2018. This procedure (testing error) is important because too flexible models could have overfitting, which means good predictions with the training dataset but bad behavior with new datasets. In this work, SMV was the model which produced better values after including new data in most of the yields, proving the goodness of the model. Only the dataset of total yield 2018 had bad behavior after including new testing data with RF and SVM models. The differences in the accuracy among different dataset tested is due to the testing set’s (10% of the samples) unbiased facts of the probability of distribution: p(x,y). Compared with previous studies in alfalfa, our work provided a methodology that notably increases the accuracy of GS prediction and helps in making breeding decisions based on genotypic data.

Other works had described different factors affecting the accuracy in GS such as SNPs density, prediction models or architecture and heritability of the traits [70,71]. In this work we found differences in the accuracies and RMSE values among different harvesting datasets. These differences can be explained as result of the phenotypic data variation during harvest time. To better understand, we performed a multiple linear regression with the mean results of GS (Pearson’s correlation, RMSE, or MAE) by harvest in parameters of broad sense heritability ($H^{2}$), residual SD, $R^{2}$, and coefficient of variation of phenotypic values (Table 3). Pearson’s correlation values of GS were not explained by the phenotypic $H^{2}$, residual SD, $R^{2}$, or coefficient of variation with a multiple $R^{2}$ of 0.588 and *p*-value: 0.069. However, we found that RMSE values of GS were correlated with residual SD, $R^{2}$, and coefficient of variation (multiple $R^{2}$ of 0.998 and *p*-value: 3.675e-07) and MAE values were correlated with residual SD, and coefficient of variation (multiple $R^{2}$ of 0.9945 and *p*-value: 1.651e-07). In the multiple correlations of RMSE and MAE residual SD was the most significant predictor variable with the effects of 1.208 and 0.929, respectively.

Our GS approach with prediction accuracy reached up to 0.793 in yield data for two years. It can be used to predict yields in the next cycles. Similarly Li et al. [72] found that total biomass yield reduces the prediction accuracy because it is necessary to have high quality phenotypic data with low residual SD in each harvest. The complex relationships among multiple traits or the same trait collected in different seasons may affect predictivity. Based on the present results of GWAS and GS, it is possible to infer that a non-additive effect may play a key role in controlling agronomic traits of alfalfa under salt stress.

## 4. Materials and Methods

### 4.1. Plant Materials

Three hundred and four alfalfa individuals from 38 half-sib families were developed by polycross with the original parents of cultivars Malone, Salado, Saranac, Alfagraze, P53V08, Renovator, Spreader III, Wrangler 5, Archer II, Cimarron, Forager, Mesa Sirsa, and U2948 2, followed by four cycles of recurrent selection for salt tolerance. Two populations, the SII and ChkSltn populations were selected based on plant survival in a greenhouse following the method described by Peel et al. [73]. In 2009, these two populations were established in a saline field nursery located near Castle Dale, UT and irrigated with high saline water. An additional cycle of selection was completed based on survival and agronomic performance, particularly forage yield under field conditions. Selected material from the two populations was then placed in a single greenhouse crossing block and combined into a single population. This material was then subjected to greenhouse screening for salt resistance as described by Peel et al. [73] and 38 plants were selected and recombined in a crossing block. The progeny from these 38 plants represent 38 half-sib families tested.

### 4.2. Phenotyping and Data Analysis

Three hundred and four individual plants from the 38 half-sib families (eight plants/family) were clonally propagated, maintaining six clones per plant in greenhouse under controlled environmental conditions. Clones from the same original plants were used for the field trails. Prior to field establishment, plot soil salinity was measured 24–48 h following a late June irrigation and averaged 7.4 dS m^−1^. Salinity of the irrigation water was also recorded and varied but was typically in the range of 7–9 dS m^−1^. Historical average annual precipitation at the site has been 20.4 cm (https://www.usclimatedata.com, 11 September 2019). In the establishment year and as part of field preparation, 70 kg ha^−1^ mono-ammonium phosphate (11N-52P-0K) was applied prior to establishing the trial providing 7.7 and 36.4 kg ha^−1^ each of N and P, respectively. Based on subsequent soil tests no other amendments were needed. A randomized complete block design with three replications was used in the field trial. One plant was grown per plot with plants on one meter spacings. Above ground fresh weight biomass (yield) was collected from the field during July, August, and September of 2018 and May, June, July, and September of 2019. Plant vigor under salt stress was scored for each plant a 1–5 scale, where 1 = weak and 5 = vigorous. Susceptible (‘AZ-90NDC-ST) and tolerant (‘AZ-88NDC’) standard checks from the Forage Production Under Salt Stress standard test were included as references [74].

Phenotypic data were spatially corrected using splines to obtain the best linear unbiased estimates (BLUEs) of fixed effects. BLUEs were estimated using a two-dimensional P-spline mixed model with Mr.Bean web application [15] using the SpATS package [75] and mixed model was defined as [76]:$$y=X\beta +f\left(r,c\right)+Z_{u}u+Z_{g}g+\varepsilon$$
where the vector $y=\left(y_{1},\ldots ,y_{304}\right)$ contains the yield in grams per plot in 304 plants, β is a vector of fixed effects including the intercept, and X is the association design matrix, f(r,c) is a smooth bivariate function of rows $r=\left(r_{1},\ldots ,r_{60}\right)$ and columns $c=\left(c_{1},\ldots ,c_{16}\right)$ corresponding to the vector of random spatial effects. u is a vector of random row and column effects accounting for discontinuous field variation with the associated matrix $Z_{u}$. g is the genotypic vector with $Z_{g}$ as the associated design matrix treated as fixed effects, and ε is the random error vector $\varepsilon =\left(\varepsilon_{1},\ldots ,\varepsilon_{304}\right)\sim N\left(0,\sigma_{\varepsilon }^{2}I_{304}\right)$. Additionally, BLUEs values for yield by year 2018 and 2019 were obtained including month (m) as random effect in the model (Appendix A):$$y=X\beta +f\left(r,c\right)+Z_{u}X+Z_{g}g+m+\varepsilon$$

Broad-sense heritability ($H^{2}$), residual standard deviation, $R^{2}$, and coefficient of variation were calculated with Mr.Bean with genotype as random factor (Table 1).

### 4.3. DNA Extraction and Sequencing

Genomic DNA was extracted from 304 original plants used for clonal propagation using a Qiagen DNEasy 96 Plant Kit (Qiagen, Valencia, CA) following the manufacturer’s instructions. DNA concentration and quality were measured using a NanoDrop ND1000 spectrophotometer (NanoDrop Technologies, Inc. Wilmington, DE). The extracted DNA was sequenced at the University of Minnesota Genomic Center for GBS according to Elshire et al. [77]. The sequencing was carried out on an Illumina HiSeq 2000 sequencer, producing single-ended reads of 100 bp each. A total of 240,444,007 reads were obtained from the population.

### 4.4. GBS and Variant Calling

The raw sequencing data (fastq files) were obtained and used for aligning to the *Medicago truncatula* genome v5.0 [13] using Bowtie2 v2.2.6 [78] with highly sensitive parameters (modified from the script S2 in [79]). Variants were called with NGSEP (Next Generation Sequencing Experience Platform) software v4.0.0 [80] and filtered at (i) maximum value allowed for a base quality score: 30; (ii) minimum allele frequency of 0.05; (iii) maintained positions at least 70% of the samples are genotyped; (iv) minimum genotyping quality 40; (v) ploidy = 4; (vi) imputation using hidden Markov model implemented in NGSEP v4.0.0. After filtering, 6862 high quality SNP markers were obtained and used in further analyses.

### 4.5. Dosage Analysis and Association Mapping

The variant call format (VCF) file with biallelic single nucleotide variants (SNVs) was transformed into GWASpoly format [5] using NGSEP software v4.0.0 [80] based on genotype field BSDP: number of base call (depth) for the all nucleotides. BDSP specify the read depth sorted as A, C, G, and T (i.e., 0,0,16,0 corresponds to GGGG, 4,0,12,0 correspond to GGGA, 8,0,8,0 correspond to AAGG) and was corroborated with the python script VCF2SM and SuperMASSA software [81] which uses Bayesian network to address allele dosage.

The association studies were performed using the R package GWASpoly using a Q+K linear mixed model as follow [5]:y=Xβ+ZSτ+ZQv+Zu+ε
where y corresponds to the observed phenotypes; β is a vector of fixed-effects; X is a incidence matrix used to model environmental effects; v is the subpopulations vector effects; Q in an incidence matrix for a population of size m; u is a polygenic effects vector; Z is a matrix of incidence mapping genotypes to observations; τ is a SNPs effects vector; S is a structure incidence matrix and ε is a residuals vector [5].

The GWAS analyses were generated with six different models including general, additive, diploidized additive, diploidized general, duplex dominant (A>B & B>A), and simplex dominant (A>B & B>A) with the dataset from BLUEs yield values. Finally markers were identified using a threshold of Bonferroni > 0.05 and they were annotated using the *M. truncatula* genome v5.0 genome browser [13].

### 4.6. Genomic Prediction

VCF file with allele dosage was numerically transformed using the python scripts VCF2SM and SuperMASSA software [81] and convert-tet-vcf.py [82]. The numerically-transformed VCF was used for GS. Eight models were tested: rrBLUP [83], BayesA, BayesB, BayesC, Bayesian ridge regression (BRR), and Bayesian LASSO (BL) from the BGLR package [84], support vector machine (SVM) from the R package Kerlab [85], and random forest (RF) from the R package Ranger [86]. For the models rrBLUP, BayesA, BayesB, BayesC, BRR, and BL the predictive ability was calculated based on 10-fold cross-validation with a training set and testing set fractions of 90% and 10% of genotypes, respectively, with the GROAN R package [87]. For the models SVM and RF the predictive ability was calculated as before, using Caret R package [88]. The predictive ability of the models was calculated as Pearson’s correlation between GEBV and phenotypes of test population, root mean squared error (RMSE), and mean absolute error (MAE). The rrBLUP assumes a lineal mixed additive model represented by the equation:$$y_{i}=X\beta +Zu+\varepsilon_{i};u\sim N\left(0,\;K\sigma_{u}^{2}\right)$$
where $y_{i}$ is a vector of observations $\left\{y_{1},\ldots ,y_{272}\right\}$, β is a vector of fixed-effects, u is a vector for genomic breeding values to follow normal distribution, X and Z are designed matrices, $\varepsilon_{i}$ is a vector of residual effects with an assumed normal distribution $\varepsilon_{i}\sim N\left(0,\sigma_{e}^{2}\right)$, and K is a positive semidefinite matrix.

The Bayesian models for continuous variables are represented by the equation:$$y_{i}=1\mu ++\overset{m}{\underset{j=1}{\sum }}X_{ij}\beta_{j}+\varepsilon_{i}$$
where $y_{i}$ is the vector of adjusted phenotypic observations $\left\{y_{1},\ldots ,y_{272}\right\}$, μ is the overall mean for the trait, $\beta_{j}$ is a vector of the marker effects associated to the columns of the marker incidence matrix, $X_{ij}$ is the *j*th SNP genotype of plant i, m is the number of markers, and $\varepsilon_{i}$ is a vector of residual effects with an assumed normal distribution $\varepsilon_{i}\sim N\left(0,\sigma_{e}^{2}\right)$.

SVM and RF are machine learning methods for classification and regression tasks [11,89]. SVM implements nonlinear regression finding a good fitting separating hyperplane. Parameters tuned up were (i) sigma (σ) (gamma for e1071 package): default = 1/(data dimension) and (ii) cost I which is cost of constrain violation = {0.25, 0.5, 1.0} with a radial kernel $\left(e^{-\sigma {\left(a-b\right)}^{2}}\right)$ to predict GEVB. RF regression was carried out using random subsamples of data and using the combined result for prediction of GEBV. Parameters tuned up were (i) mtry: number SNPs of randomly selected at each tree node {2, 116, 6832}. For regression models, the number of predictor variables split at in each node (rounded down), and (ii) splitting rule were used during tree construction for regression “variance” or “extra-trees” with a node-size = 5.

## 5. Conclusions

Marker–trait association identified a group of 27 SNP markers associated with salt tolerance. BLAST search in the reference genome revealed several functional genes associated with the significant marker loci and assigned as putative candidate genes based on their roles in response to salt stress. Additionally, genomic selection allowed to predict the breeding values on Logan alfalfa population for salt tolerance with good accuracy. Among the models tested, the machine learning methods were the best models according to high Pearson’s correlation and low RMSE values in yields of different harvests and vigor under salt stress for two years. The identification of the models and the accuracies obtained in this work are likely sufficient to predict breeding values in breeding programs for salt tolerance in alfalfa.

## Figures and Tables

**Figure 1 ijms-21-03361-f001:**
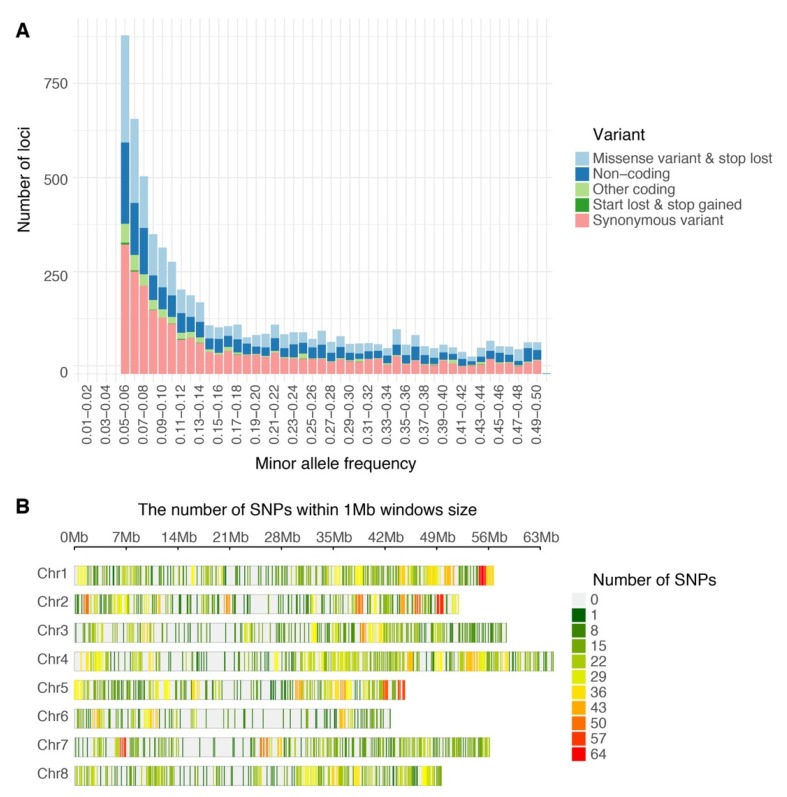
Single nucleotide polymorphism variants (SNVs) identified in alfalfa (*Medicago sativa*) populations developed in Logan, Utah (**A**) Histogram of filtered variants called by Next Generation Sequencing Experience Platform (NGSEP) showing distribution by minor allele frequency and classified by function after annotation. (**B**) Distribution of GBS SNP markers across eight *Medicago truncatula* chromosomes using 1 Mb window. The colored lines represent the marker density as showing on the right color legends.

**Figure 2 ijms-21-03361-f002:**
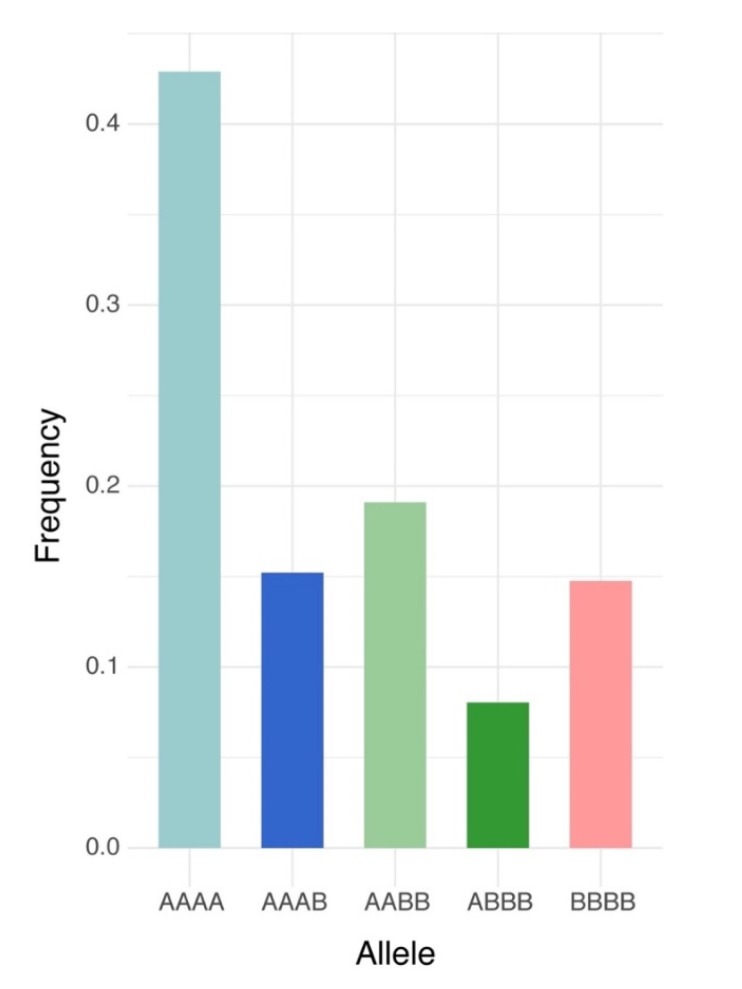
Frequency of allele dosage in autotetraploid alfalfa (*Medicago sativa*) for 6862 high-quality biallelic SNVs obtained from NGSEP pipeline in the Logan dataset. A represents dosage of the major allele and B is for the minor allele dosage.

**Figure 3 ijms-21-03361-f003:**
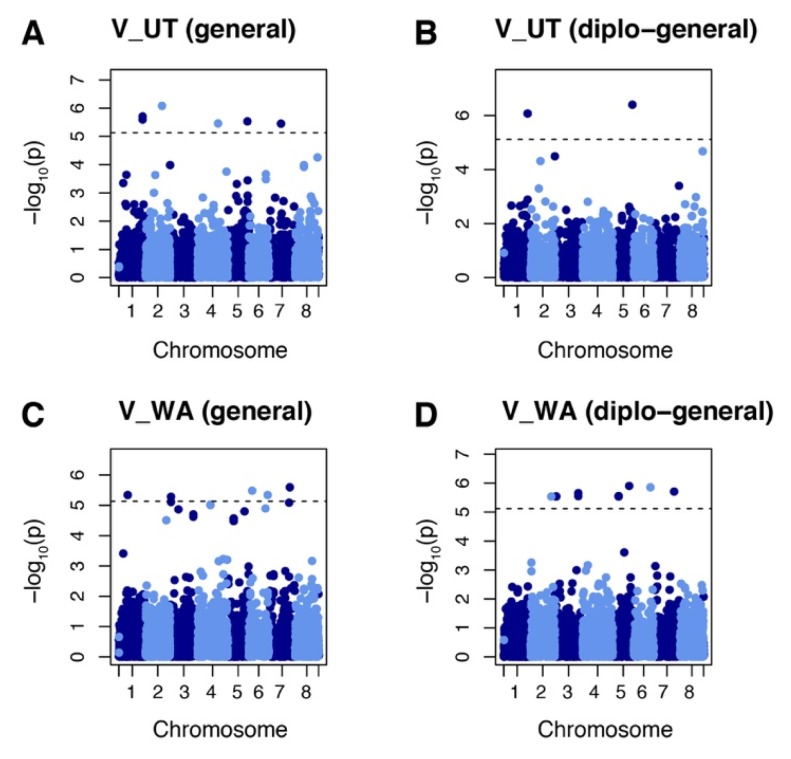
Manhattan plots showing marker–trait association for vigor (V) in alfalfa populations at Othello Washington (WA) and Castle Dale Utah (UT). (**A**) Markers identified by general model in the UT dataset. (**B**) Markers identified by diplo-general model in the UT dataset. (**C**) Markers identified by general model in the WA dataset. (**D**) Markers identified by diplo-general model in the WA dataset. The threshold of 0.05 was used for significant markers according to the Bonferroni method.

**Figure 4 ijms-21-03361-f004:**
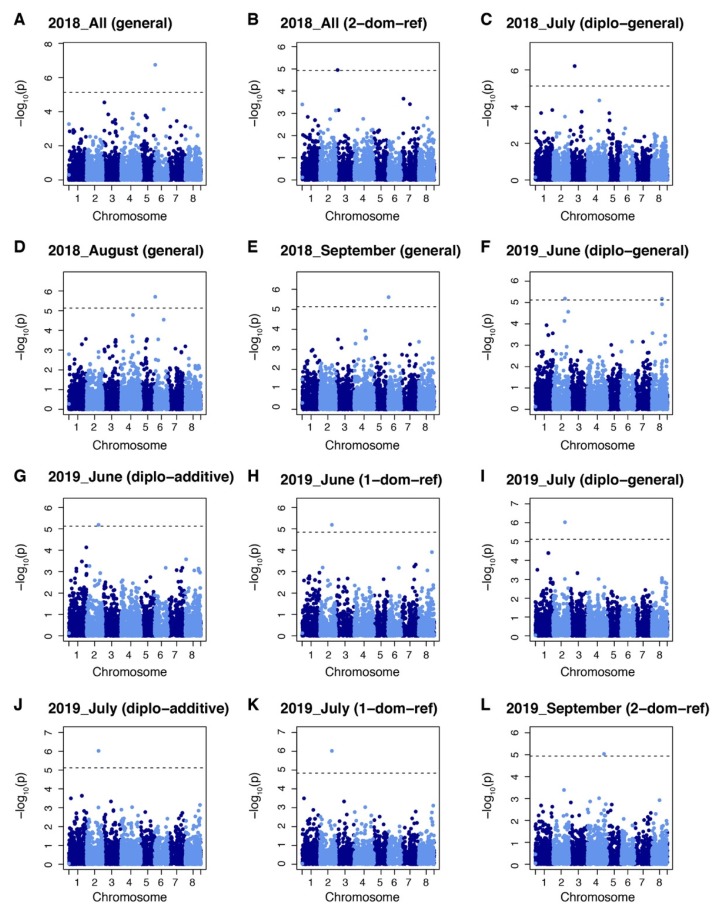
Manhattan plots showing marker–trait associations for yield datasets in alfalfa (*Medicago sativa*) at Othello, Washington over two years. (**A**) Markers identified by general model in All 2018. (**B**) Markers identified by 2-dominant reference model in All 2018. (**C**) Markers identified by diplo-general model in July 2018 dataset. (**D**) Markers identified by general model in August 2018. (**E**) Markers identified by general model in September 2018. (**F**) Markers identified by diplo-general model in June 2019. (**G**) Markers identified by diplo-additive model in June 2019. (**H**) Markers identified by 1-dominant reference model in June 2019. (**I**) Markers identified by diplo-general model in July 2019. (**J**) Markers identified by diplo-additive model in July 2019. (**K**) Markers identified by 1-dominant reference model in July 2019. (**L**) Markers identified by 2-dominant reference model in September 2019 dataset. Markers threshold was set using Bonferroni > 0.05.

**Figure 5 ijms-21-03361-f005:**
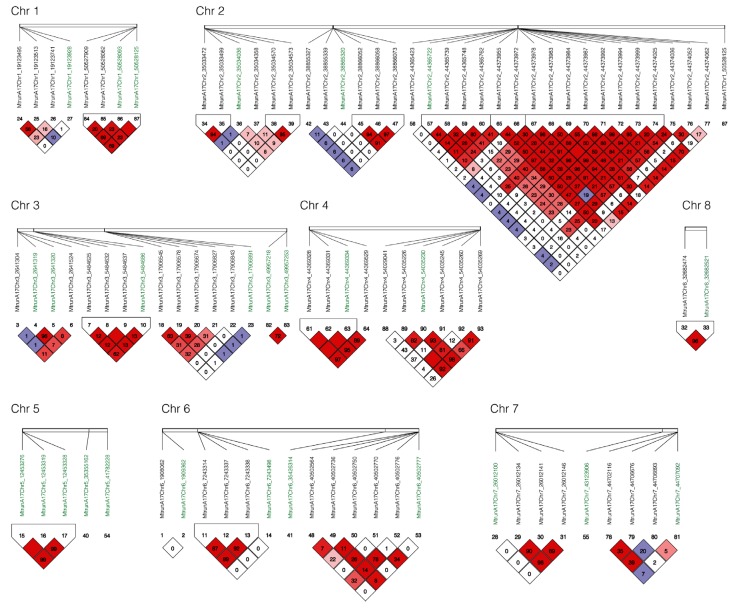
Linkage disequilibrium (LD) among markers associated for yield and vigor under salt stress. Haploview v4.2 [14] and pairwise LD values ($r^{2}\times 100$) were used for 27 SNPs associated with yield and vigor under salt stress (green color) and their surrounding SNPs in 10 kb (black color). Bright red coloring indicates $D^{\prime }=1,LOD\geq 2$; blue coloring indicates $D^{\prime }=1,LOD<2$; white coloring indicates $D^{\prime }<1,LOD<2$; shades of pink/red coloring indicates $D^{\prime }<1,LOD\geq 2$.

**Table 1 ijms-21-03361-t001:** A summary of single nucleotide polymorphism (SNP) markers developed by genotype-by-sequencing (GBS) and their categories of gene annotations based on the *Medicago truncatula* reference genome (*Mt*.v5.0).

SNPs		Count
Coding	Synonymous variant	2843
	Missense variant	2014
	Stop lost	3
	Stop gained	22
	Start lost	0
	Splice donor variant	2
	Splice acceptor variant	4
	Exonic splice region variant	7
	Splice region variant	83
	5 prime UTR variant	82
	3 prime UTR variant	174
Non-coding	Upstream transcript variant	61
	Downstream transcript variant	32
	Intron variant	956
	Intergenic variant	579

**Table 2 ijms-21-03361-t002:** SNP marker, trait, model, chromosome, position, allele, −logp, locus tag, and putative gene function associated with alfalfa (*Medicago sativa*) yield (Y) under salt stress in Othello, WA, and vigor (V) in Othello, Washington (V_WA), and Castle Dale, Utah (V_UT) fields.

M.	Trait	Model	Chr.	Position	SNP	−logp	Locus tag	Annotation
283	V_WA	1	1	19123928	A/G	5.34	MtrunA17_Chr1g0170381	Hypothetical protein
860	V_UT	1, 2	1	50528093	C/T	5.59, 6.08	MtrunA17_Chr1g0205221	Putative folate-biopterin transporter, major facilitator superfamily domain-containing protein
861	V_UT	1, 2	1	50528125	C/T	5.7, 6.08		
1561	V_UT	1	2	35034036	A/G	6.08	MtrunA17_Chr2g0312131	Hypothetical protein
1644	Y_Jun_19, Y_Jul_19	2, 3, 5	2	38865320	A/G	5.19, 6.02	MtrunA17_Chr2g0316741	Hypothetical protein
1744	V_WA	2	2	44365722	A/G	5.54	MtrunA17_Chr2g0324021	Putative oxidoreductase
1992	V_WA	2	3	2641319	C/G	5.55	MtrunA17_Chr3R0014140	RLX_singleton_family134 PWWP domain
1993	V_WA	1, 2	3	2641320	C/T	5.28, 5.53		
2033	Y_All_18	4	3	5484686	C/G	4.95	MtrunA17_Chr3g0083861	Putative Serpin family protein
2195	Y_Jul_18	2	3	17906891	C/T	6.2	MtrunA17_Chr3g0094791	Putative tetratricopeptide-like helical domain, DYW domain-containing protein
2711	V_WA	2	3	49957218	A/T	5.65	NA	NA
2712	V_WA	2	3	49957253	C/T	5.55		
3515	V_UT	1	4	44369334	C/T	5.46	MtrunA17_Chr4g0048811	Putative aminoacyltransferase, E1 ubiquitin-activating enzyme
3708	Y_Sep_19	4	4	54035230	A/G	5.04	MtrunA17_Chr4g0062111	Putative protein CHAPERONE-LIKE PROTEIN OF POR1
4154	V_WA	2	5	12453276	A/G	5.55	MtrunA17_Chr5g0410771	Putative HSP20-like chaperone, P-loop containing nucleoside triphosphate hydrolase
4155	V_WA	2	5	12453319	G/T	5.55		
4156	V_WA	2	5	12453328	C/G	5.54		
4463	V_WA	2	5	35355162	G/T	5.91	MtrunA17_Chr5g0435221	Putative 23S rRNA (adenine(2503)-C(2))-methyltransferase
4633	V_UT	1, 2	5	41782228	A/T	5.53, 6.4	MtrunA17_Chr5g0444321	Putative leucine-rich repeat domain, L domain-containing protein
4775	Y_All_18, Y_Aug_18, Y_Sep_18	1	6	1909362	C/T	6.74, 5.7, 5.61	MtrunA17_Chr6g0451341	Putative transcription regulator IWS1 family
4868	V_WA	1	6	7243498	A/G	5.48	MtrunA17_Chr6g0457561	Hypothetical protein
5146	V_WA	2	6	35426314	C/G	5.86	MtrunA17_Chr6R0226110	Putative potassium channel, voltage-dependent, ERG
5241	V_WA	1	6	40502777	A/G	5.34	MtrunA17_Chr6g0486011	Putative zinc finger, RanBP2-type
5558	V_UT	1	7	26012100	C/T	5.45	MtrunA17_Chr7g0235641	Putative RIN4, pathogenic type III effector avirulence factor Avr cleavage
5834	V_WA	2	7	43123906	A/G	5.71	NA	NA
5858	V_WA	1	7	44707092	C/T	5.6	MtrunA17_Chr7g0259771	Putative small GTPase superfamily, EF-hand domain pair
6478	Y_Jun_19	2	8	32682521	A/T	5.18	MtrunA17_Chr8g0369441	Putative brevis radix (BRX) domain, transcription factor BREVIS RADIX domain-containing protein

M. = Marker consecutive. Chr. = chromosome; Y = BLUEs values for yield in the indicated harvest; HS = health score of plants under salt stress. Models: 1 = general, 2 = diplo-general, 3 = diplo-additive, 4 = 2-dominant-reference, 5 = 1-dominant-reference. Locus tag annotation based on [13]. Orange colored cells indicate the same marker in different traits. Grey colored cells indicate several markers associated to same loci.

**Table 3 ijms-21-03361-t003:** Genomic selection (GS) metrics for alfalfa (*Medicago sativa*) plant vigor under salt stress at Castle Dale, Utah (HS_UT), and Othello, Washington (HS_WA). Eight GS models were tested using 10-fold cross-validation and the metrics of accuracies as Pearson’s correlation values (Pearson) and root mean squared error (RMSE) are shown by model.

Dataset	Metric	rrBLUP	BayesA	BayesB	BayesC	BL	BRR	RF	SVM
V_UT	Pearson	0.267	0.274	0.250	0.275	0.272	0.245	0.244	0.287
	RMSE	0.894	0.885	0.896	0.890	0.887	0.894	0.890	0.880
V_WA	Pearson	0.336	0.336	0.327	0.342	0.329	0.343	0.324	0.361
	RMSE	0.696	0.693	0.698	0.692	0.696	0.696	0.708	0.691

Notes: BL, Bayesian LASSO; BRR, Bayesian ridge regression; RF, random forest; SVM, support vector machine.

**Table 4 ijms-21-03361-t004:** Description of best linear unbiased estimates (BLUEs) yield values and genomic selection (GS) results for alfalfa (*Medicago sativa*) grown under salt stress. Broad sense heritability ($H^{2}$), residual SD (Res_SD), $R^{2}$, and coefficient of variation (Coef_Var) of phenotypic data were calculated using the package Mr.Bean [15] with genotype as random effect. Eight GS models were tested using 10-fold cross-validation and the metrics of accuracies as Pearson’s correlation values (Pearson) and root mean squared error (RMSE) are shown by model.

Dataset	$H^{2}$	Res_SD	$R^{2}$	Coef_Var	Metric	rrBLUP	BayesA	BayesB	BayesC	BL	BRR	RF	SVM
Jul_18	0.47	0.55	0.479	0.23	Pearson	0.305	0.305	0.303	0.307	0.303	0.299	0.343	0.324
					RMSE	0.509	0.506	0.51	0.508	0.508	0.509	0.508	0.503
Aug_18	0.51	0.46	0.51	0.25	Pearson	0.27	0.259	0.275	0.272	0.253	0.265	0.268	0.24
					RMSE	0.409	0.411	0.407	0.408	0.408	0.408	0.414	0.414
Sep_18	0.69	0.24	0.629	0.38	Pearson	0.444	0.445	0.448	0.447	0.454	0.45	0.464	0.509
					RMSE	0.255	0.254	0.254	0.255	0.254	0.254	0.256	0.244
All_18	0.8	0.38	0.717	0.3	Pearson	0.234	0.216	0.227	0.226	0.209	0.236	0.302	0.268
					RMSE	0.377	0.38	0.376	0.379	0.375	0.377	0.37	0.371
May_19	0.43	0.55	0.506	0.28	Pearson	0.116	0.108	0.107	0.121	0.119	0.115	0.182	0.113
					RMSE	0.551	0.558	0.556	0.552	0.552	0.553	0.541	0.548
Jun_19	0.33	0.5	0.502	0.28	Pearson	0.173	0.147	0.155	0.146	0.184	0.154	0.219	0.201
					RMSE	0.477	0.481	0.478	0.478	0.474	0.478	0.467	0.469
Jul_19	0.43	0.49	0.555	0.33	Pearson	0.258	0.242	0.238	0.266	0.231	0.235	0.287	0.281
					RMSE	0.51	0.513	0.509	0.507	0.51	0.51	0.514	0.51
Sep_19	0.54	0.29	0.553	0.39	Pearson	0.249	0.231	0.257	0.24	0.247	0.236	0.276	0.301
					RMSE	0.31	0.312	0.309	0.311	0.309	0.31	0.312	0.308
All_19	0.83	0.37	0.716	0.45	Pearson	0.072	0.065	0.083	0.064	0.06	0.083	0.137	0.138
					RMSE	0.464	0.467	0.466	0.466	0.463	0.462	0.456	0.455

Notes: BL, Bayesian LASSO; BRR, Bayesian ridge regression; RF, random forest; SVM, support vector machine.

**Table 5 ijms-21-03361-t005:** Comparison of genomic selection (GS) models in phenotypic data collected for alfalfa (*Medicago sativa*) yield under salt stress. Random forest (RF) and support vector machine (SVM) models were trained by 10-fold cross validation (RF_10CV or SVM_10%). Pearson’s correlation (Pearson) and root mean squared error (RMSE) values were calculated.

Harvest	Metric	RF_10CV	RF_10%	SVM_10CV	SVM_10%
July_2018	Pearson	0.343	0.728	0.324	0.793
	RMSE	0.508	0.389	0.503	0.353
August_2018	Pearson	0.268	0.225	0.240	0.279
	RMSE	0.414	0.468	0.414	0.459
September_2018	Pearson	0.464	0.771	0.509	0.729
	RMSE	0.256	0.222	0.244	0.205
All_2018	Pearson	0.302	0.259	0.268	-0.073
	RMSE	0.370	0.399	0.371	0.657
May_2019	Pearson	0.182	0.135	0.113	0.282
	RMSE	0.541	0.511	0.548	0.491
June_2019	Pearson	0.219	0.226	0.201	0.353
	RMSE	0.467	0.479	0.469	0.464
July_2019	Pearson	0.287	0.365	0.281	0.479
	RMSE	0.514	0.471	0.510	0.450
September_2019	Pearson	0.276	0.410	0.301	0.627
	RMSE	0.312	0.302	0.308	0.275
All_2019	Pearson	0.137	0.275	0.138	0.229
	RMSE	0.456	0.469	0.455	0.472

## Data Availability

BLUE values of biomass fresh weight are presented in Appendix A. The row data of GBS were submitted to the NCBI Sequence Read Archive with bioproject ID: PRJNA611554 and biosample # SAMN14336867.

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
