# Peer review of "Genome-Wide Association and Prediction of Traits Related to Salt Tolerance in Autotetraploid Alfalfa (Medicago sativa L.)"

_ijms, 2020, doi:10.3390/ijms21093361_

Round 1

Reviewer 1 Report

The authors adequately satisfied my comments and improved their manuscript as well. I feel confortable in suggesting the manuscript as suitable for publication in the present form

Author Response

Thanks

Reviewer 2 Report

This is a valuable manuscript regarding genome-wide association and traits improvement during salt tolerance in legume plant Alfalfa. The manuscript is relatively well-performed, and result corroborates with previous research. However, some flaws that can be corrected to improve the understanding the manuscript and will recommend for minor revision.

 Line 76-79. This form of salt stress therefore affects plants ….. and Line 80-84. Mechanisms of plant resistance to this form. Theses line seems very long and complicated, hence need to modify.

Please check the quality and visibility of Figure 1 and Figure 4. Kindly improve the quality of mentioned both figures.

Author Response

Responses to reviewer #2’s Comments and Suggestions

This is a valuable manuscript regarding genome-wide association and traits improvement during salt tolerance in legume plant Alfalfa. The manuscript is relatively well-performed, and result corroborates with previous research. However, some flaws that can be corrected to improve the understanding the manuscript and will recommend for minor revision.

Response:  We improved it by correction the flaws throughout the manuscript.

 Line 76-79. This form of salt stress therefore affects plants ….. and Line 80-84. Mechanisms of plant resistance to this form. Theses line seems very long and complicated, hence need to modify.

Response: We rewrote these parts and shorten the sentences (L52-59).  

Please check the quality and visibility of Figure 1 and Figure 4. Kindly improve the quality of mentioned both figures.

Response: We improved the quality of these figures and replaced them.

This manuscript is a resubmission of an earlier submission. The following is a list of the peer review reports and author responses from that submission.

Round 1

Reviewer 1 Report

This is a valuable manuscript regarding traits improvement and salt tolerance in important legume Alfalfa plant. The manuscript is well written and result corroborates with previous research. Thus, I feel the paper is appropriate for publication in the present form.

Author Response

This is a valuable manuscript regarding traits improvement and salt tolerance in important legume Alfalfa plant. The manuscript is well written and result corroborates with previous research. Thus, I feel the paper is appropriate for publication in the present form.

Response: Thank you for your comments

Reviewer 2 Report

The paper of Hawkins and collaborators use a set of informatic tools in order to determine the main markers-trait association and genomic selection related with salt tolerance in plants of alfalfa. Although I do not have any problem with the different informatic approaches the authors use, I think the paper has a problem with the source of data that is used for the different determinations. The different sets of data from different field locations/green-house experiment are not used for all the different determinations. As far as I can understand from the manuscript, phenotypic variation is based on data from two field areas and the green-house experiment, and the physiological parameters measured at the field and at the green house experiment are different. For the association mapping analyses, the author use two field experiments, but one field experiment and the green house one are missing. DNA extraction and sequencing is done in green-house plants only, with no data on the field experiment. Under my point of view, the data used for this time of analyses need to be similar and all used for the different determinations. I do not see the point of using certain experiments for only certain determinations. The data need to be all complete from the different experimental units to have strong and convincing results. Furthermore, there is not information on the amount of salt and nutrients from the field experiments, that in principle are set-up in salt areas, nor weather conditions for the experimentation. The same occurs for the soil used for the green house experiment. Also, the experiment on salt stress at the green house is done with 100 mM NaCl, and there is not information if that amount resemble any of the field conditions.

The authors should rethink the way the data is presented, if they only want to present field data, or green-house data, or all the data together, but always comparing the same parameters and the same plots for all the determinations.

Author Response

The paper of Hawkins and collaborators use a set of informatic tools in order to determine the main markers-trait association and genomic selection related with salt tolerance in plants of alfalfa. Although I do not have any problem with the different informatic approaches the authors use, I think the paper has a problem with the source of data that is used for the different determinations. The different sets of data from different field locations/green-house experiment are not used for all the different determinations. As far as I can understand from the manuscript, phenotypic variation is based on data from two field areas and the green-house experiment, and the physiological parameters measured at the field and at the green house experiment are different.

Response: The data sets we used are from three independent experiments. Two field experiments were performed in UT and WA for obtaining biomass under salt stress. They were done by two different groups and each group focused on their favorite traits for better understanding the genetic base of salt tolerance in the same population. The greenhouse experiment was to measure physiological factors in the controlled environmental conditions. We realized that it would be better to measure same traits for all experiments. However, some of the trait such as LCC and SC are difficult to obtained in the field due to the uncontrolled environment conditions. To avoid genetic variations among different lines, we used clonally propagated cuttings form the same original plants for all three experiments. We believe that these experiments could be complementary to each other.

For the association mapping analyses, the author use two field experiments, but one field experiment and the green house one are missing.

Response: Yes. the association mapping result for greenhouse has been published separately as cited in the manuscript (Liu et al. 2019). AM results for two field experiments were presented in Fig. 2.

DNA extraction and sequencing is done in green-house plants only, with no data on the field experiment.

Response: We used clones from original plants for both field and greenhouse experiments.  DNA was extracted from the original plants and subjected to sequencing.

Under my point of view, the data used for this time of analyses need to be similar and all used for the different determinations. I do not see the point of using certain experiments for only certain determinations. The data need to be all complete from the different experimental units to have strong and convincing results.

Response: We agree. We had some difficulty to measure physiological traits in field due to too much variation in the field as we were unable to control open environment conditions. We therefore used greenhouse experiment for measuring those traits. We measured biomass in both field and greenhouse conditions.

Furthermore, there is not information on the amount of salt and nutrients from the field experiments, that in principle are set-up in salt areas, nor weather conditions for the experimentation. The same occurs for the soil used for the green house experiment. Also, the experiment on salt stress at the green house is done with 100 mM NaCl, and there is not information if that amount resemble any of the field conditions.

Response: we added this information in the Material and Methods under subtitle “Phenotyping and Data Analysis” on page 7. We used 100 mM NaCl since this concentration showed better result for salt stress on alfalfa plants in greenhouse pots.

The authors should rethink the way the data is presented, if they only want to present field data, or green-house data, or all the data together, but always comparing the same parameters and the same plots for all the determinations.

Response: We agree. We had some difficulty to measure physiological traits in field due to too much variation in the field conditions.

Reviewer 3 Report

The manuscript aim at evaluating genomic-wide association of traits related to salt tolerance in Medicago sativa L. The authors reported results of genetic analyses for eight traits related to salt tolerance in an alfalfa breeding population with three different phenotypic datasets.

The manuscript aim has a high scientific appeal.The science is sound, the manuscript well-written, clear and concise. Discussion is adequate to the results provided by the authors. Just a few minor comments are reported below:

-Line 53-69: some references should be added to support those statements (for example Manuela Chaves' group or Rana Munns' group papers are excellent group about plant salinity). Also reference to salinity have to be updated to more recent literature.

-why the concentration of 100 mM NaCl was selected?

-Table 1: GH meaning has to be explained

-Discussion and conclusion: It should be nice to describe more in detail (deeper than that stated in line 321-337) in which way the marker loci associated to salinity can explain their involvement from a physiological point of view.

Author Response

The manuscript aim at evaluating genomic-wide association of traits related to salt tolerance in Medicago sativa L. The authors reported results of genetic analyses for eight traits related to salt tolerance in an alfalfa breeding population with three different phenotypic datasets.

The manuscript aim has a high scientific appeal. The science is sound, the manuscript well-written, clear and concise. Discussion is adequate to the results provided by the authors. Just a few minor comments are reported below:

-Line 53-69: some references should be added to support those statements (for example Manuela Chaves' group or Rana Munns' group papers are excellent group about plant salinity). Also reference to salinity have to be updated to more recent literature.

Response: We added references from both groups and cited their work in the introduction.

-why the concentration of 100 mM NaCl was selected?

Response: We used 100 mM NaCl since this concentration showed better result for salt stress on alfalfa plants in greenhouse pots.

-Table 1: GH meaning has to be explained.

Response: We added the explanation for GH

-Discussion and conclusion: It should be nice to describe more in detail (deeper than that stated in line 321-337) in which way the marker loci associated to salinity can explain their involvement from a physiological point of view.

Response: We added more discussion in this section on pages 16-17.

(x) Moderate English changes required.

Response: We had a native language scientist editing English language for the manuscript as mentioned in the ACKNOWLEDGEMENTS.

Round 2

Reviewer 2 Report

The authors have included some changes into the manuscript but I still think the experimental design is not appropriate, with different groups working in different field and green-house conditions, different growth conditions and different determinations for each experimental unit. Under my point of view, the paper has not improved too much from the last version. I am sorry but I cannot recommend it for publication.